# Golgi Phosphoprotein 3 Regulates the Physical Association of Glycolipid Glycosyltransferases [note 1]

**DOI:** 10.3390/ijms231810354

**Published:** 2022-09-08

**Authors:** Fernando M. Ruggiero, Natalia Martínez-Koteski, Viviana A. Cavieres, Gonzalo A. Mardones, Gerardo D. Fidelio, Aldo A. Vilcaes, Jose L. Daniotti

**Affiliations:** 1Centro de Investigaciones en Química Biológica de Córdoba (CIQUIBIC), Consejo Nacional de Investigaciones Científicas y Técnicas (CONICET), Universidad Nacional de Córdoba, Córdoba X5000HUA, Argentina; m.natalia.martinez@unc.edu.ar (N.M.-K.); gfidelio@unc.edu.ar (G.D.F.); 2Departamento de Química Biológica Ranwel Caputto, Facultad de Ciencias Químicas, Universidad Nacional de Córdoba, Córdoba X5000HUA, Argentina; 3Facultad de Medicina y Ciencia, Universidad San Sebastián, Valdivia 5110466, Chile; viviana.cavieres@uss.cl (V.A.C.); gonzalo.mardonesc@uss.cl (G.A.M.); 4CIQUIBIC (UNC-CONICET), Departamento de Química Biológica, Facultad de Ciencias Químicas, Universidad Nacional de Córdoba, Córdoba X5000HUA, Argentina; daniotti@fcq.unc.edu.ar

**Keywords:** glycosyltransferases, ST3Gal-II, β3GalT-IV, GOLPH3, glycolipids, gangliosides, Golgi apparatus, glycosyltransferase complex, glycosylation

## Abstract

Glycolipid glycosylation is an intricate process that mainly takes place in the Golgi by the complex interplay between glycosyltransferases. Several features such as the organization, stoichiometry and composition of these complexes may modify their sorting properties, sub-Golgi localization, enzymatic activity and in consequence, the pattern of glycosylation at the plasma membrane. In spite of the advance in our comprehension about physiological and pathological cellular states of glycosylation, the molecular basis underlying the metabolism of glycolipids and the players involved in this process remain not fully understood. In the present work, using biochemical and fluorescence microscopy approaches, we demonstrate the existence of a physical association between two ganglioside glycosyltransferases, namely, ST3Gal-II (GD1a synthase) and β3GalT-IV (GM1 synthase) with Golgi phosphoprotein 3 (GOLPH3) in mammalian cultured cells. After GOLPH3 knockdown, the localization of both enzymes was not affected, but the fomation of ST3Gal-II/β3GalT-IV complex was compromised and glycolipid expression pattern changed. Our results suggest a novel control mechanism of glycolipid expression through the regulation of the physical association between glycolipid glycosyltransferases mediated by GOLPH3.

## 1. Introduction

Together with glycoproteins and glycosaminoglycans, glycolipids are an important part of the glycosylation repertoire of cells. They are mainly located on the outer leaflet of the plasma membrane where they regulate several physiological processes [1]. Additionally, qualitative and quantitative changes in the glycosylation of glycolipids are a hallmark of tumors contributing to their development and progression [2,3,4]. Thus, the aberrant glycolipid expression reported in tumors is emerging as an attractive target to deliver specific therapies [2,5]. These molecules are synthesized by a complex membrane-bound machinery formed by glycolipid glycosyltransferases (GGTs) at the Golgi complex [6,7] (Figure 1). The GGTs have the typical type II membrane protein topology with an N-terminal domain (NTD) composed of the cytoplasmic tail, a transmembrane domain (TMD) and a short luminal stem region, followed by C-terminus that bears the catalytic domain [8]. The length and volume of the TMD, the continuous cycle between distal and proximal Golgi compartments and the amino acid motifs present at the NTD are some of the molecular features that promote the retention and localization of GGTs to specific sub-Golgi compartments [9,10,11,12]. In addition, the association of some GGTs, mediated by the catalytic domain and/or the N-terminal domain, constitutes homo-and heterocomplexes at the Golgi apparatus that can affect the enzyme localization and activity [13,14,15,16,17,18]. In spite of the advanced knowledge on the key role of glycosylation in health and disease, the molecular basis underlying mechanistic details of glycosylation and the players involved in this process are not properly understood.

Golgi phosphoprotein 3 (GOLPH3) was the first Golgi-associated oncoprotein to be reported [19]. Its gene copy number and protein expression levels are increased in several solid tumors [20]. GOLPH3 overexpression mediates tumorigenesis by a sustained and enhanced mTOR signaling [21,22,23] and provides a prognostic biomarker of tumor progression [24,25,26]. GOLPH3 has been implicated in several cellular processes [25], including vesicle trafficking [27,28], maintenance of Golgi distribution [27], cytokinesis [29] and response to DNA damage [30]. Despite its importance in cancer biogenesis, the role of GOLPH3 in the regulation of glycolipid metabolism remains an underexplored area of research which only recently gained attention. As mentioned above, glycosyltransferases stay dynamically concentrated in Golgi membranes. While a group of these proteins do not require GOLPH3 for Golgi localization [27,31,32], others, such as some GGTs, need the presence of GOLPH3 to control its sub-Golgi localization and rate of lysosomal degradation [11,33,34]. Thus, GOLPH3 oncogenic properties might partly be mediated by its role in the regulation of GGTs and cellular glycosylation.

In the present work, we explored the participation of GOLPH3 in the metabolism of complex sialyl-glycolipids, namely GD1a and GM1 gangliosides. We found that both ST3Gal-II and β3GalT-IV glycosyltransferases (GD1a and GM1 synthase, respectively) are physically associated. The association of these GGTs depends on GOLPH3 and modulates the biosynthesis of glycolipids. Our results suggest a novel function for GOLPH3 in the organization of multienzymatic glycosyltransferase complexes in the Golgi and in the control of cellular glycosylation.

## 2. Results

### 2.1. Knockdown of GOLPH3 Is Associated with Changes in the Distribution of the Golgi Complex

To study the involvement of GOLPH3 in the metabolism of glycolipids, we took advantage of the T98G cell line, a broadly used model of glioblastoma multiforme [35,36]. In addition, T98G cells show high levels of GOLPH3 in respect to control human astrocytes from primary culture, indicating that this cell is useful for studying the functional effects of lowering the expression of GOLPH3 [37,38]. First, we asked if the downregulation of GOLPH3 was associated with changes in the morphology of the Golgi complex, the main organelle in which glycolipid synthesis occurs. To this end, we employed a T98G cells with a 90% reduction in the expression of GOLPH3 (GOLPH3 knockdown KD cells; Figure 2A,B) [39]. Consistent with the literature, a detailed confocal microscopy analysis revealed abnormal changes in the Golgi complex distribution after GOLPH3 KD (Figure 2C). Briefly, cells were fixed, permeabilized and incubated with specific antibodies that recognize the endogenous expression of two Golgi-resident proteins, namely, polypeptide GalNAc-transferase-2 (ppGNT-2) and the cis-Golgi matrix protein of 130 kDa (GM130). Finally, cells were probed with Alexa Fluor-conjugated secondary antibodies. Results showed that both, ppGNT-2 and GM130, localized to the Golgi complex of cells with different GOLPH3 expression; however, the pattern of distribution is strikingly different (Figure 2C). In GOLPH3 KD cells, the Golgi complex showed a decreased extension in the xy-plane (Figure 2C,D), while keeping its volume constant (Figure 2E) and increasing its extension in the z-plane (Figure 2C). These results further support the idea that GOLPH3 has a crucial role in the global organization of the Golgi complex [27,30,31,33,40,41].

### 2.2. GOLPH3 Knockdown Is Associated with Changes in Glycolipid Expression

Alterations in the structure of the Golgi complex were previously associated with deficits in the synthesis, glycosylation and/or trafficking of proteins and lipids. Specifically, various pharmacological treatments were found to mediate Golgi disorganization resulting in the inhibition of complex ganglioside expression [42,43,44]. More recently, changes in glycolipid expression on the cell surface were reported following Golgi complex disorganization by the knockout of the Golgi reassembly-stacking proteins 1 and 2 (GRASP65 and GRASP55, respectively), both essential for Golgi structure formation and function [16]. Taking these antecedents into account, we asked whether changes in the Golgi complex distribution resulted in a modified glycolipid expression in T98G cells (see glycolipid biosynthesis summary in Figure 1). We first analyzed the ganglioside composition by high-performance thin layer chromatography (HPTLC). Results showed a drastic change in the ganglioside expression pattern after GOLPH3 knockdown (Figure 3A). In particular, GOLPH3 KD cells exhibited a downregulation of ganglioside GD1a with a concomitant increase in gangliosideGM1 (Figure 3A, arrowheads).

To further confirm these results, we next examined the GM1 and GD1a ganglioside levels by flow cytometry and confocal microscopy through the binding of cholera toxin and a monoclonal antibody, respectively [45]. In agreement with HPTLC results, downregulation of GD1a with GM1 upregulation was observed in T98G GOLPH3 KD cells (Figure 3B–D). Additionally, knocking down of GOLPH3 in another tumor-related cell line, MCF7, led to similar changes in GM1 and GD1a gangliosides (Figure 3E,F) suggesting that GOLPH3 may have a general role in glycolipid metabolism in multiple tumor types. In addition, the Golgi morphology upon GOLPH3 knockdown in MCF7 is similar to T98G GOLPH3 KD cells (Appendix A). It is worth mentioning that the specificity of the labeling was confirmed by using a genetically modified CHO-K1 cell line (CHO-K1 GM1+/GD1a+) (positive control). The parental cell line (CHO-K1 GM1-/GD1a-) mainly expresses GM3 and was used as negative control (Figure 3G,H). Collectively, our results show a downregulation of GD1a with a concomitant GM1 upregulation after GOLPH3 KD supporting a role for the Golgi-associated GOLPH3 protein in the surface expression of complex sialyl-glycolipids.

### 2.3. GOLPH3 Expression Does Not Affect the Subcellular Localization or N-Glycosylation of ST3Gal-II Sialyltransferase

We next sought to understand the mechanism linking GOLPH3 with the changes in GD1a and GM1 expression. In the Golgi apparatus, the complex pattern of gangliosides is generated by the stepwise addition of sugars catalyzed by specific glycosyltransferases (Figure 1) [6]. GM1 is generated via addition of N-Acetylgalatosamine (GalNAc) to GM2 by β-1,3-galactosyltransferase 4 (β3GalT-IV). GD1a is then synthesized by addition of a sialic acid to GM1 by the enzyme β-Galactoside α-2,3-Sialyltransferase 2 (ST3Gal-II) [10,46]. To test if the subcellular localization of these enzymes is modulated by GOLPH3, we performed a confocal microscopy analysis of cells transiently expressing the amino terminus of ST3Gal-II fused to mCherry (ST3Gal-II^(1−51)^-mCherry) [13] and a fusion protein of β3GalT-IV containing amino acids 1–52 and the HA epitope fused to YFP (β3GalT-IV^(1−52)^-HA-YFP) [21] (Figure 4). It is worth mentioning that the NTD of these enzymes is necessary to retain them in the Golgi complex (reviewed in [6]). Depleting GOLPH3 did not affect the subcellular distribution of both enzymes, since ST3Gal-II and β3GalT-IV showed similar colocalization levels with the cis-Golgi marker GM130 in control and GOLPH3 KD cells (Figure 4A–D). Our results indicate that the retention of these enzymes at the Golgi complex do not depend on GOLPH3 levels. Therefore, the GOLPH3-mediated changes in surface expression of sialyl-glycolipids may not be attributed to the mislocalization of these enzymes.

GOLPH3 has been shown to modulate O- and N-glycosylation events [47,48,49,50] and we have also previously demonstrated that N-glycosylation of ST3Gal-II is required for proper enzyme localization in the Golgi complex of CHO-K1 cells and for its appropriate enzymatic activity [10]. We next explored whether N-glycosylation of ST3Gal-II was affected in cells expressing low levels of GOLPH3. In CHO-K1 cells, full length ST3Gal-II was mainly expressed as a 43 kDa polypeptide that is heavily N-glycosylated at Asn^211^ (Figure 4E and [10]). Prevention of ST3Gal-II N-glycosylation by site-directed mutagenesis of Asn^211^ (ST3Gal-II N211Q) or by blockade of polypeptide synthesis by the general glycosylation inhibitor tunicamycin, caused the reduction in its molecular mass from 43 to 41 kDa, clearly showing that ST3Gal-II is N-glycosylated at Asn211 in CHO-K1 cells (Figure 4E and [10]). ST3Gal-II was also N-glycosylated in T98G control as well as in GOLPH3 KD cells, since it was expressed as a 43 kDa polypeptide (Figure 4E). Thus, these results indicate that depletion of GOLPH3 does not affect N-glycosylation of ST3Gal-II.

Some GGTs, not only require the presence of one or more N-glycans for acquiring a proper folding and thus Golgi localization and activity, but also an appropriate processing of the N-glycan while they transit through the Golgi complex [10,15]. This prompted us to evaluate the quality of the N-glycan present on ST3Gal-II under different GOLPH3 expression levels. The successive processing of the oligosaccharide by the cis/medial Golgi resident N-acetylglucosaminyl transferase-I and mannosidase-II confers endoglycosidase H (Endo-H) resistance and terminal glycosylation of glycoproteins N-glycans. Treatment of ST3Gal-II with Endo-H caused a reduction in its molecular mass from 43 to 41 kDa in both T98G control and GOLPH3 KD cells (Figure 4E), indicating that the N-glycan on ST3Gal-II contained a high proportion of mannose residues and that most of the enzyme did not progress beyond the medial Golgi compartment, in agreement with the sub-Golgi localization described by confocal microscopy (Figure 4A–D). Altogether, these results strongly suggest that GOLPH3 expression does not influence the N-glycosylation or the N-glycan composition of ST3Gal-II in T98G cells, and therefore, the downregulation of GD1a observed in GOLPH3 KD cells may not be attributable to a lack of posttranslational modifications of this enzyme.

### 2.4. Role of GOLPH3 in the Physical Association of ST3Gal-II and β3GalT-IV Ganglioside Glycosyltransferases

The association of ganglioside glycosyltransferases constitutes homo- and heterocomplexes at the Golgi apparatus that can affect both, enzyme localization and activity [13,14,15,16,17,18,44,51]. It has been demonstrated that N-terminal domain of β4GalNAcT-I (GM2 synthase) and β3GalT-IV is relevant for the interaction between them. The formation of a physical and functional complex between β4GalNAcT-I and β3GalT-IV, determines the efficient conversion of GM3 to GM1 [51]. In this way, a channeling of substrates occurs and the product of the first enzyme is preferentially used by the second one and not by a competing transferase. This was also observed for ST3Gal-V (GM3 synthase) and ST8Sia-I (GD3 synthase) complex in which lactosylceramide is efficiently guided through the complex to produce GM3 and GD3 [13]. As shown in Figure 3D–F, GOLPH3 KD cells have a drastic reduction in the expression of GD1a (ST3Gal-II product) with a concomitant increase in the expression of GM1 (ST3Gal-II substrate). Thus, this prompted us to ask whether there is a physical association between β3GalT-IV and ST3Gal-II responsible of enhancing the conversion of GM1 into GD1a, and the participation of GOLPH3 in the formation of this complex. In order to assess these questions, cells expressing ST3Gal-II^(1−51)^-mCherry and β3GalT-IV^(1−52)^-HA-YFP were processed for co-immunoprecipitation followed by Western blot (Figure 5A). In T98G control cells, ST3Gal-II and β3GalT-IV co-immunoprecipitated showing for the first time that these two enzymes are part of the same complex. Moreover, depletion of GOLPH3 leads to a reduction in the physical association of ST3Gal-II and β3GalT-IV (Figure 5A), suggesting that GOLPH3 modulates the formation and/or maintenance of the complex in which the N-terminal domains are involved. The putative association of GOLPH3 with β3GalT-IV and ST3Gal-II was assessed by immunoprecipitation and Western blotting. Our results showed that GOLPH3 interact with both, the truncated form of β3GalT-IV (β3GalT-IV^(1−52)^-HA-YFP) and the full length version of ST3Gal-II (ST3Gal-II-HA) (Figure 5B–C).

As a complementary approach, and to characterize the interaction between both glycosyltransferases in vivo, we then performed a Förster Resonance Energy Transfer (FRET) analysis. For these experiments, the YFP- and CFP-tagged versions of ST3Gal-II (ST3Gal-II^(1−51)^-EYFP) and β3GalT-IV (β3GalT-IV^(1−52)^-CFP) were coexpressed in T98G and GOLPH3 KD cells. The FRET efficiency for this pair of glycosyltransferases was maximal (~1) in control cells (Figure 5D, left panel and F). However, FRET efficiency was significantly reduced in GOLPH3 KD cells (Figure 5D, right panel and F), and comparable to the FRET efficiency of soluble versions of coexpressed YFP and CFP (non-interacting proteins), (Figure 5E,F). These results indicate that the association between ST3Gal-II and β3GalT-IV in the Golgi complex is impaired when GOLPH3 levels are reduced.

Taken together, our results show a previously uncharacterized role for GOLPH3 in the metabolism of glycolipids, namely, the modulation of glycosyltransferases association at the Golgi complex. The loss of physical interaction between ST3Gal-II and β3GalT-IV after GOLPH3 knockdown underlies the changes in cell surface ganglioside pattern. In this way, the establishment and maintenance of the GOLPH3/glycosiltransferases complex at the Golgi complex might be crucial to fuel the synthesis of GD1a through the channeling of GM2 and GM1 substrates.

## 3. Discussion

Cell surface glycosylation has key roles in a myriad of biological processes [2,52]. Therefore, understanding the mechanisms involved in its regulation is fundamental in our effort to better comprehend physiological and pathological cellular states and to treat disease. GOLPH3 has been found to interact with Golgi-resident glycoprotein–glycosyltransferases in mammalian cells influencing cell-surface glycosylation patterns, and consequently cell function, by controlling Golgi localization of the corresponding enzymes [34,49,50,53]. Here, we show that GOLPH3 alter the glycolipid expression of human glioblastoma (T98G) and breast cancer (MCF7) cell lines. When GOLPH3 levels decrease, GD1a is downregulated with a concomitant GM1 upregulation at the cell surface. The de novo synthesis of GM1 and GD1a is carried out in the Golgi complex by the sequential addition of galactose and then a sialic acid residue on GM2 by the GGTs β3GalT-IV and ST3Gal-II, respectively (Figure 1). The majority of GGTs selectively localize to specific sub-Golgi compartments. They have the typical type II transmembrane protein topology, with a N-terminal domain (NTD) constituted by a short cytoplasmic tail, a transmembrane region and a luminal short stem region, followed by a globular catalytic C-terminal domain [8]. The NTD is critical for the proper Golgi localization of GGTs in mammalian cells [6,54]. The dramatic change in Golgi morphology after GOLPH3 down regulation (Figure 2) [27], opened the possibility that GOLPH3 modulates the localization of ST3Gal-II and β3GalT-IV. However, our data show that the retention of these enzymes at the Golgi complex do not depend on GOLPH3 levels. Moreover, GOLPH3 expression does not influence the N-glycosylation of ST3Gal-II, a crucial post-translational modification for proper enzyme localization in the Golgi complex and for its appropriate enzymatic activity [34]. In agreement with other reports, the Golgi-mediated localization of mammalian glycosyltransferases by GOLPH3 seems not to be a general phenomena since some of these enzymes did not require GOLPH3 to reside at the Golgi complex [27,31]. Furthermore, changes in cell surface syalilation were reported after GOLPH3 knockdown without affecting Golgi localization of syaliltransferases [55]. Therefore, the role of GOLPH3 in the regulation of glycosylation is not restricted to its ability to mediate the retention of glycosyltransferases in the Golgi complex [34,49,50,53]. Rather, GOLPH3 might regulate cell surface glycosylation by other mechanisms.

A large number of studies have demonstrated that glycosyltransferases are able to form multienzyme complexes with each other [8,18,44,51,56]. In particular, at least two complexes of ganglioside glycosyltransferases with participation of their N-terminal domains have been described in Chinese Hamster Ovary (CHO)-K1 cells, one formed by β4GalT-VI, ST3Gal-V and ST8Sia-I [18,44], mainly of proximal Golgi localization, and the other one formed by β3GalT-IV and β4GalNAcT-I, of more distal Golgi localization [51]. Here, we show for the first time a physical association between β3GalT-IV and ST3Gal-II by co-immunoprecipitation experiments from membranes of T98G cells expressing epitope-tagged versions of these enzymes (NTD-fused reporter tags). Furthermore, the FRET microscopy technique allowed us to confirm the in situ occurrence of an interaction between β3GalT-IV and ST3Gal-II N-terminal domains at the Golgi complex. These results indicate, as for the case of the other complexes mentioned above, that the N-terminal domains are involved in the interactions between β3GalT-IV and ST3Gal-II enzymes. A number of biochemical and cell biological studies have provided convincing evidence for the existence of glycosyltransferase complexes that improve the enzymatic activity of one of the partners [13,44,51]. As a consequence, a disruption of the complex formed by β3GalT-IV and ST3Gal-II, emerges as a plausible hypothesis to explain the downregulation of GD1a observed in GOLPH3 KD cells. Using fluorescence microscopy and biochemical techniques we provide, to the best of our knowledge, the first evidence that GOLPH3 interacts with and modulates the formation of a complex including β3GalT-IV and ST3Gal-II glycosyltransferases. This physical interaction underlies the changes observed in the surface expression pattern of gangliosides in cancer cells. In this way, the establishment and maintenance of the GOLPH3/glycosiltransferases complex may be crucial to fuel the synthesis of GD1a through the channeling of GM2 and GM1 substrates. Recently, we demonstrated that the gangliosides pattern in lipopolysaccharide-stimulated macrophages showed an increment of GD1a with a concomitant decrease in GM1, product and substrate of ST3Gal-II, respectively [57]. In addition, an increase om GOLPH3 in tumor-associated macrophages (TAMs) has been observed [58]. These observations suggest that GOLPH3 can also participate in the regulation of gangliosides expression in macrophages. Further studies are needed to verify this premise.

Pioneer work from laboratory of Professor Maccioni [6,59] supports the idea that the transmembrane domains (TMDs) are involved in the associations between glycosyltranferases. Recently, our laboratory described that some glycosyltranferases are S-acylated at conserved cysteine residues located close to the cytoplasmic border of their TMDs [8]. In the case of ST3Gal-II, its NTD is able to form homotypic covalent dimers through a unique cysteine residue located in its cytoplasmic tail. However, β3GalT-IV does not present any cysteine residue in the cytoplasmic/TMD regions, which can engage in such interactions. Thus, non-covalent interactions arise as possible contributor to the nature of the interaction between β3GalT-IV and ST3Gal-II. In addition, GOLPH3 emerges as a potential and crucial partner in the formation of the GGT complex by bringing both enzymes in close proximity in the first instance and by providing a favorable environment for the interaction between the NTDs. In this sense, the interaction between GOLPH3 and some glycosyltransferases through their cytosolic tails has been suggested [34]. More research is needed to better understand the biochemical nature association between these GGTs and GOLPH3 as well as the relative binding affinity between the three partners.

Despite the current knowledge regarding the formation of GGTs complexes, the mechanisms regulating glycosylation in both health and disease remain unclear. Several features such as the organization, stoichiometry and composition of these complexes may impact their sorting properties, sub-Golgi localization and enzymatic activity. Here, we describe a new GGT complex between β3GalT-IV and ST3Gal-II. Furthermore, we show that GOLPH3 plays a crucial role in the formation of this GGT complex and by doing so, it influences the glycolipids profile that human glioblastoma and breast cancer cells express at the cell surface. This novel level of regulation of glycan synthesis opens up new questions to explore the molecular characterization of other complexes, where the oncoprotein GOLPH3 may also operate and participate in glycosylation pathways as well.

## 4. Materials and Methods

### 4.1. Cell Lines, Transfection and Electroporation

T98G, MCF7 and CHOK1 cells (ATCC, Manassas, VA, USA) were maintained at 37 °C, 5% CO2 in DMEM (Dulbecco’s modified Eagle’s medium) supplemented with 10% (*v*/*v*) FBS and antibiotics (100 μg/mL penicillin and 100 μg/mL streptomycin). T98G GOLPH3 knockdown cells were a generous gift from Prof. Gonzalo Mardones (Department of Physiology, School of Medicine, Universidad Austral de Chile, Valdivia, Chile and Center for Interdisciplinary Studies of the Nervous System (CISNe), Universidad Austral de Chile, Valdivia, Chile) [38]. For confocal microscopy experiments, cells were transfected with 1 μg/35-mm-diameter dish of the indicated plasmid using 25 kDa linear polyethyleneimine (PEI) (Polyciences, Inc., Warrington, PA, USA) (PEI:DNA, 2:1) and allowed for 24 h of protein expression. For immunoprecipitation experiments, 2 × 10^6^ cells per condition were resuspended in BTX Disposable Cuvettes Model #640 (4 mm gap) with 100 µL of electroporation mix (80 µL of solution I plus 20 µL of solution II) containing 2 µg of the indicated plasmid and then pulsed in a BTX Electro Cell Manipulator 600 (voltage: 500 V, resistance: 186 Ω, capacitance: 75 µF). Solution I: 125 mM Na_2_HPO_4_; 12.5 mM KCl, adjusting pH to 7.75 with acetic acid. Solution II: 55 mM MgCl_2_. Both solutions were prepared in water, filtered and kept at −20 °C until use. After electroporation, cells were seeded in a 60-mm-diameter dish and allowed for 24 h of protein expression.

For GOLPH3 knockdown assays in MCF7 cells, 1 × 10^6^ cells were resuspended in BTX Disposable Cuvettes Model #640 (2 mm gap) with 100 uL of electroporation mix containing 3 µg of each shRNA against GOLPH3 plasmids (mentioned below) and then pulsed (voltage: 155 V, resistance: 186 Ω, 950 µF). After electroporation, cells were grown on Lab-Tek II chambered coverglass (Thermofisher, Waltham, MA, USA) for ~30 h.

### 4.2. Cloning

The sequences of shRNA against GOLPH3 (5′-TCTGGATTACGTGGCTGTATGTTAAT CAAGAGTTAACATACAGCCACGTAATCCAGA-3′ and 5′-GGAGTGTCTGAAGGCC AATACTCAAGAGGTATTGGCCTTCAGACACTCC-3′) were inserted into L307 lentiviral vector (gift of Dr. Ege T. Kavalali, Vanderbilt University). The electroporation was performed as described above.

### 4.3. Immunoprecipitation and Western Blot

Cells grown on a 60-mm-diameter dish for 24 h were harvested and then lysed during 60 min on ice with 300 µL of lysis buffer (50 mM Tris-HCl; pH 7.2; 1.0% Triton X-100; 300 mM NaCl; 1mM PMSF; protease inhibitor cocktail). Lysates were centrifuged at 500g for 1 min to remove DNA and cell debris. A fraction (10%) of the resulting supernatant (input) was kept and the rest was treated with anti-HA mAb (1:75 dilution) for 90 min on ice. Then, 50 µL of protein G-Sepharose beads (Amersham Pharmacia, 75% suspension washed 3× with lysis buffer before use) were added and incubated overnight on a rotating wheel at 4 °C. Immunocomplexes were pelleted by centrifugation at 12,000× *g* for 30 s at 4°C, the supernatant was kept (flow through) and the beads were then washed three times at 4 °C with lysis buffer. After that, 40 µL of Laemmli sample buffer (Bio-rad Laboratories, Hercules, CA, USA) supplemented with 5% 2-mercaptoethanol was added to protein G-Sepharose beads and 10 µL of sample buffer was added to the input sample and to 10% of the flow through sample. Finally, all samples were heated at 95 °C for 3 min and centrifuged at 12,000× *g* for 2 min to pull down protein G-Sepharose beads. Proteins were resolved by electrophoresis through 12% SDS-polyacrylamide gels under reducing conditions and then were electrophoretically transferred to nitrocellulose membranes for 60 min at 350 mA. Nonspecific binding sites on the nitrocellulose membrane were blocked for 60 min with 5% (*w*/*v*) non-fat dried milk in PBS, and incubated overnight at 4 °C with primary antibodies produced in rabbit diluted in PBS (1:2000 anti-GFP; 1:2000 anti-HA; 1:3000 anti-RFP). After three washes with PBS, membranes were incubated with secondary antibodies diluted in PBS (1:10,000 goat polyclonal antibody to mouse IgG (IRDye 800CW, LI-COR)) for 60 min at room temperature. Bands of proteins were detected using an Odyssey infrared imaging system according to the manufacturer’s protocols (LI-COR Biotechnology, Lincoln, NE, USA). Molecular masses were calculated based on calibrated standards ran in parallel.

### 4.4. Confocal Immunofluorescence Microscopy

For cell surface ganglioside labeling, cells were grown on Lab-Tek II chambered coverglass (Thermofisher), and incubated on ice for 10 min. Then, an appropriate dilution of anti-GD1a antibody and Alexa Fluor 647-conjugated cholera toxin subunit B was added and incubated for 45 min on ice. After that, cells were washed three times with DMEM, fixed with 4% (*w*/*v*) paraformaldehyde in PBS for 10 min at 4 °C and permeabilized with 0.1% Triton X-100/200 mM glycine in PBS for 2 min at room temperature. Finally, cells were incubated with 1:10,000 dilution of Hoechst dye to stain DNA and 1:1000 dilution of Alexa Fluor 488-conjugated goat anti-mouse IgG secondary antibody. Confocal images were collected using an Olympus Fluoview FV-1000 and Olympus Fluoview FV-1200 laser-scanning confocal microscope equipped with an argon/helium/neon laser. Single confocal sections of 0.8 μm were taken parallel to the coverslip (xy sections) with a 63× and 1.4 numerical aperture objective lens. Images were acquired and processed with the FV10 lsm image software and FIJI software (NIH). Final images were compiled with Adobe Photoshop CS6. The fluorescence micrographs shown are representative of at least three independent experiments. For 3D reconstruction of the Golgi complex and colocalization experiments, full z-stacks of this organelle were taken with a minimum voxel resolution of 41 nm at 1024 × 1024.

### 4.5. FRET Analysis

Cells were grown at 80% confluence in a Lab-Tek II chambered coverglass (Thermofisher) and transfected to coexpress ST3Gal-II^(1−51)^–YFP and β3GalT-IV^(1−52)^-CFP as indicated above. Double CFP and YFP transfectants (cytosolic expression) were used as a negative FRET control. The temperature was maintained at 37 °C throughout the experiment using a Stage Top Incubator (Tokai Hit, Fujinomiya, Japan). Confocal images were collected using an Olympus Fluoview FV-300 laser-scanning confocal microscope. An argon laser source was used to excite the donor (β3GalT-IV^(1−52)^-CFP) and acceptor (ST3Gal-II^(1−51)^–YFP) at 458 nm and 514 nm, respectively. Emission filters of 470–500 nm and 530–570 nm bandpass were used to detect CFP and YFP fluorescence, respectively. Resulting images were processed using ImageJ software. To calculate the FRET efficiency based on the sensitized emission method [60], background values were first calculated from non-transfected cells and then subtracted from each channel. Then, the donor and acceptor spectral bleed-through values were obtained from single transfected β3GalT-IV^(1−52)^-CFP and ST3Gal-II^(1−51)^–YFP cells, respectively, and subtracted from the FRET channel. Finally, mean FRET efficiency values at the Golgi complex were calculated in a cell-by-cell and pixel-by-pixel basis. The resulting image was then pseudocolored according to the calculated FRET efficiencies.

### 4.6. Flow Cytometry

For cell surface ganglioside analysis by flow cytometry, cells were grown on 100-mm-diameter dish, harvested using trypsin and then resuspended in 100 µL of DMEM. Approximately 2 × 10^5^ cells per condition were incubated on ice for 10 min to inhibit intracellular transport. After that, 100 µL of DMEM containing anti-GD1a mAb (1:30 dilution) and Alexa Fluor 647-conjugated cholera toxin subunit B (1:15,000 dilution) was added and cells were incubated on ice for 30 min. Then, cells were washed three times with DMEM and resuspended in 200 µL of DMEM containing a 1:500 dilution of Alexa Fluor 488-conjugated secondary antibody (for anti-GD1a labeling). Final washing steps were carried out before resuspension of cells in 50 µL of DMEM. Appropriate negative (CHO-K1 cells, GM1 and GD1a negative cells) and positive controls (CHO-K1 GM1+/GD1a+, a cell line that express both, GD1a and GM1) were included in the analysis and were processed in parallel. Samples were analyzed using a FACSCanto II cytometer (BD Biosciences, San Jose, CA, USA). For each condition, forward light scatter, side light scatter, Alexa Fluor 488 and Alexa Fluor 647 fluorescence were evaluated using FlowJo software (FlowJo LLC, Ashland, OR, USA). A gate was applied in the forward scatter versus side scatter dot plot to restrict the analysis to intact cells. Doublet exclusion was performed by plotting the height against the area for forward scatter. For gated cells, the final histograms of fluorescence were evaluated. Statistical analysis and graphic representations were conducted using GraphPad prism software.

### 4.7. Plasmids and Antibodies

The molecular cloning and characterization of the expression plasmids used in this work were as follows. Plasmids coding for ST3Gal-II (uniprot ID: Q16842): ST3Gal-II-HA, ST3Gal-II-HA (N211Q) and ST3Gal-II^(1−51)^–mCherry (cDNA coding for the first 51 amino acids from the N-terminal domain (cytosolic tail, transmembrane domain, and few amino acids of the stem region) of ST3Gal-II fused to cherry fluorescent protein) [10]. To generate ST3Gal-II^(1−51)^–YFP, which drives the expression of the first 51 amino acids from ST3Gal-II fused to enhanced yellow fluorescent protein, the corresponding cDNA fragment of ST3Gal-II was subcloned into pEYFP plasmid (Clontech, Kyoto, Japan) [10]. Plasmids coding for β3GalT-IV (Q9Z0F0): β3GalT-IV^(1−52)^-YFP-HA (cDNA coding for the first 52 amino acids from the N-terminal domain (cytosolic tail, transmembrane domain, and few amino acids of the stem region) of β3GalT-IV fused to enhanced yellow fluorescent protein (YFP) and HA-tag [51] and of β3GalT-IV^(1−52)^-CFP (cDNA coding for the first 52 amino acids from the N-terminal domain (cytosolic tail, transmembrane domain, and few amino acids of the stem region) of β3GalT-IV fused to enhanced cyan fluorescent protein (CFP) [51].

We used the following monoclonal antibodies produced in mouse models: clone 16B12 to HA-tag (Biolegend), clone 35 to GM130 (BD Biosciences, San Jose, CA, USA), clone 1b7 to gangliosides GD1a/GT1b/GD1aα, a gift from P.H.H. Lopez (INIMEC-CONICET7-UNC) [61], clone DM1A to α-tubulin (Sigma-Aldrich; cat# T9026, St. Louis, MO, USA). We used polyclonal antibodies to the following proteins: ppGNT-2 (Sigma-Aldrich; cat# HPA011222), GFP (Thermofisher; cat# A-6455), HA-tag antibody (Sigma; cat# H6908), Red fluorescent protein (Sigma; cat# AB356483), GOLPH3 (Abclonal; cat# A13121; Woburn, MA, USA). The following fluorochrome-conjugated antibodies and reagents were from Life Technologies: Alexa Fluor 647 and Alexa Fluor 546-conjugated cholera toxin subunit B (cat# C34778, #C34777), Alexa Fluor 488-conjugated goat anti-mouse IgG (cat# A-10680), Alexa Fluor 546-conjugated goat anti-rabbit IgG (cat# A-11035). The following fluorochrome-conjugated antibody was from LI-COR: IRDye 800CW goat polyclonal antibody anti-mouse IgG and IRDye 680RD goat polyclonal antibody anti-rabbit IgG.

### 4.8. Thin Layer Chromatography (TLC)

Lipid extraction and chromatography were performed mostly as previously described [62].

Cells in culture were washed with phosphate-buffered saline (PBS) and harvested from the dishes with a cell scraper. After centrifugation, the pellets of cells were measured for the wet weight and were pooled for lipid extraction. Lipids fromT98G control (433 mg of wet weight) and GOLPH3 KD (371 mg of wet weight) were extracted with chloroform:methanol (2:1, *v*/*v*) at 4 °C for 24 h and centrifuged. The supernanatant was collected and the pellet subjected to a second lipid extraction by adding chloroform:methanol:water (30:60:8) at 4 °C for 24 h and centrifuged. Both supernanatants were mixed and subjected to two Folch partitions, first by adding 0.2 vol. of water and then 0.2 vol. of methanol:water (1:1). The resulting upper phases were freed from water-soluble contaminants by being passed through a Sephadex G-18 column. The lipid extract was used for thin-layer chromatography (TLC) analysis, supplemented with the appropriate amounts of standard gangliosides, and chromatographed on high performance TLC plates (HPTLC; Merck) using C:M:0.2% CaCl2 (60:36:8 *v*/*v*) as solvent. The bands were visualized by resorcinol-HCl spray and heating at 100 °C for 10 min.

### 4.9. Statistical Analyses

Results are presented as mean ± S.E.M. Statistical analyses were carried out using Student’s two-tailed unpaired t test with Graph Pad Prism 9.4.0 and were attributed at the 95% level of confidence (* *p* < 0.05 and ** *p* < 0.01, *** *p* < 0.0001).

### 4.10. Image Processing

Final images were compiled with Adobe Illustrator CC 23.0.1, with the confocal fluorescence micrographs in the present work being representative of at least three independent experiments. Scale bars in all figures represent 10 μm.

## Figures and Tables

**Figure 1 ijms-23-10354-f001:**
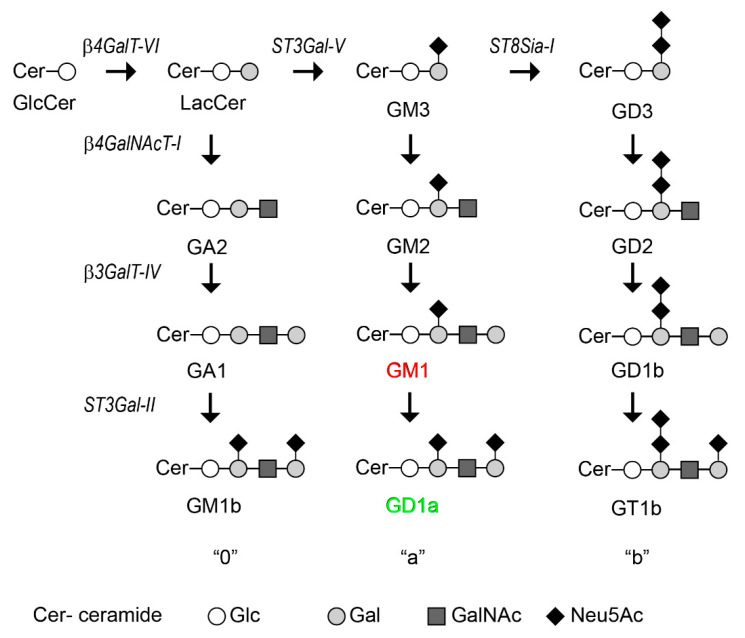
Synthesis pathway of 0-a-and b-series gangliosides by the successive stepwise addition of monosaccharides to the growing oligosaccharide chain. β4GalT-VI, UDP-Gal:glucosylceramide galactosyltransferase; ST3Gal-V, CMP-NeuAc:lactosylceramide sialyltransferase; ST8Sia-I, CMP-NeuAc:GM3 sialyltransferase; β4GalNAcT-I, UDP-GalNAc: LacCer/GM3/GD3 N-acetylgalactosaminyltransferase; β3GalT-IV, UDP-Gal:GA2/GM2/GD2 galactosyltransferase (GM1 synthase); ST3Gal-II, CMP-NeuAc:GA1/GM1/GD1b sialyltransferase (GD1a synthase). Gangliosides evaluated in the current study, GM1 and GD1a, are highlighted in red and green, respectively. Cer: ceramide; Glc: glucose; Gal: galactose; GalNAc: N-acetylgalactosamine; Neu5Ac: N-Acetylneuraminic acid.

**Figure 2 ijms-23-10354-f002:**
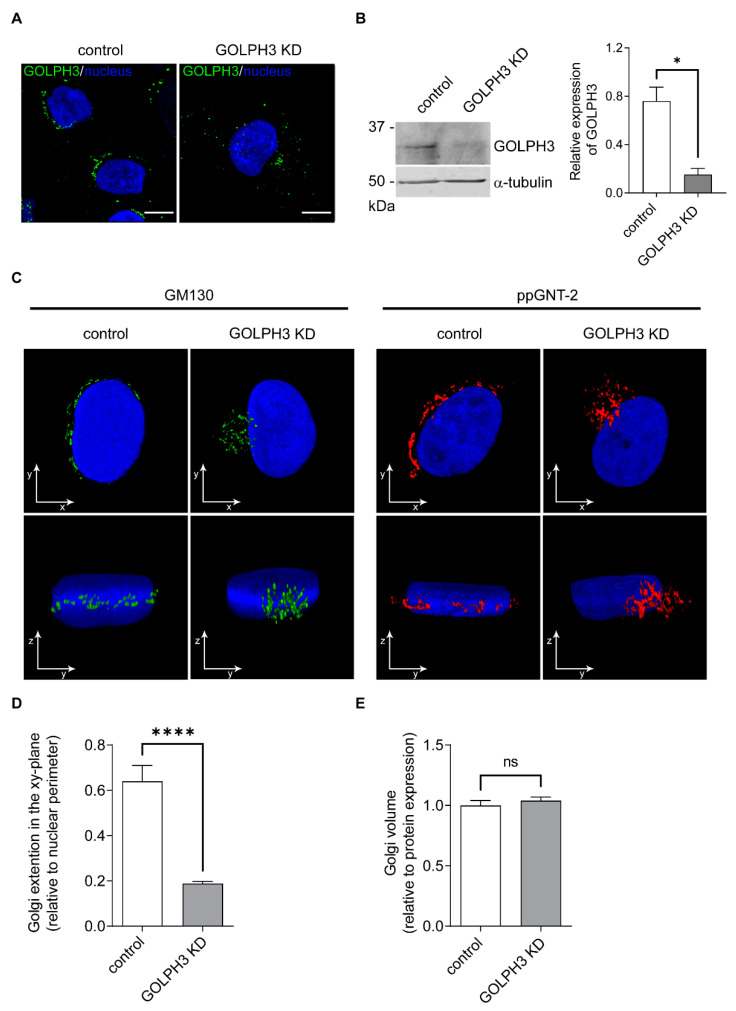
Effect of GOLPH3 on the morphology of the Golgi complex. The expression of GOLPH3 was analyzed by immunostaining (**A**) and Western blot (**B**) in T98G control and GOLPH3 KD cells. Scale bars in A: 10 µm. (**C**) Three-dimensional confocal microscopy reconstructions showing the subcellular distribution of two endogenously-expressed Golgi-resident proteins in T98G control and GOLPH3 KD cells (GM130: 130 kDa cis-Golgi matrix protein and ppGNT-2: polypeptide N-acetyl galactosaminyltransferase 2). Different views (xy-and yz-planes) of a complete Golgi reconstruction of the same cells are shown. Hoechst dye was used to counterstain the cell nucleus. (**D**) Golgi complex extension in the xy-plane was quantified and expressed relative to the nuclear perimeter (arbitrarily assigned as 1). (**E**) Golgi complex volume obtained from full reconstructions of the organelle by confocal microscopy and normalized to protein expression (fluorescence intensity). Two-tailed, unpaired t tests were carried out to assess statistical significance of results (* *p* < 0.05; **** *p* < 0.0001; ns: not significant).

**Figure 3 ijms-23-10354-f003:**
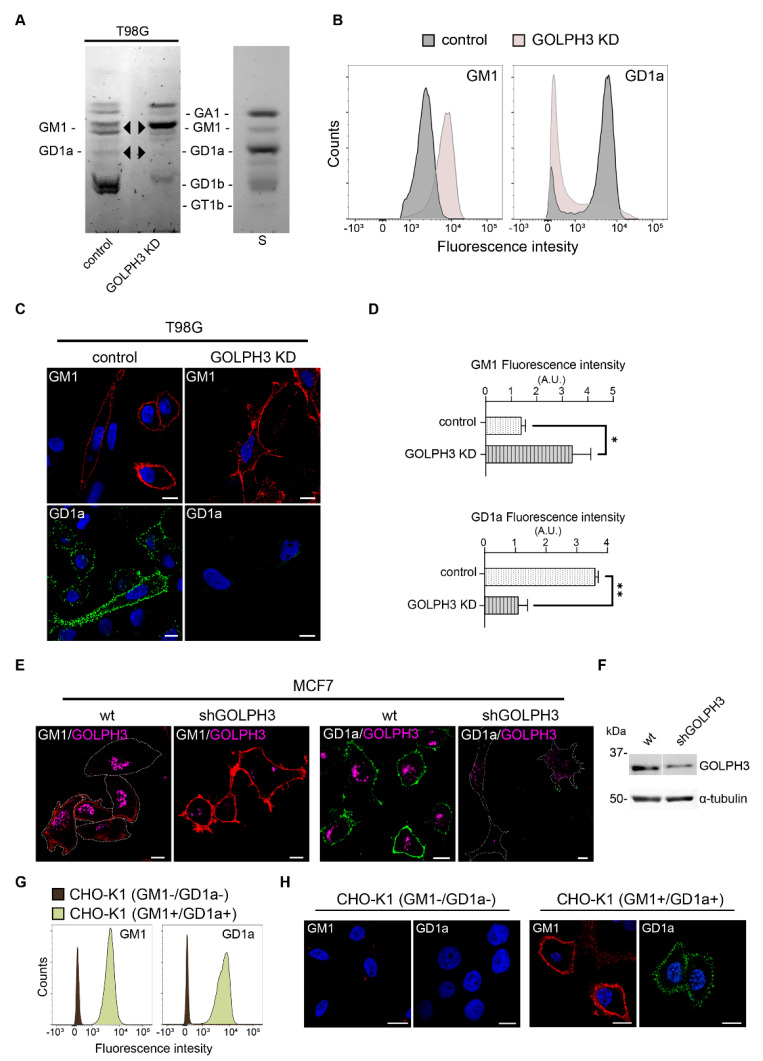
Changes in ganglioside expression associated with GOLPH3 knockdown. (**A**) High-performance thin-layer chromatography (HPTLC) analysis of ganglioside expression in T98G control (from left to right, lane 1) and GOLPH3 KD (knockdown) (lane 2) cells. Glycolipid standards (lane 3) were co-chromatographed and are indicated on the right. (**B**–**D**) T98G control and GOLPH3 KD cells were probed with anti-GD1a monoclonal antibody or with cholera toxin (GM1) and then subjected to flow cytometry analysis (**B**) and confocal microscopy (**C**). Quantification of GM1 and GD1a gangliosides, in T98G control and GOLPH3 KD cells subjected to immunofluorescence (**D**). Two-tailed, unpaired t tests were carried out to assess statistical significance of results (* *p* < 0.05; ** *p* < 0.01). A.U.: arbitrary units. (**E**,**F**) Immunostaining showing the endogenous expression of GOLPH3, GM1 and GD1a (**E**), and Western blot showing GOLPH3 levels in MCF7 control and GOLPH3 knockdown cells (**F**). Scale bars in microscopy images: 10 µm. (**G**,**H**) CHO-K1 (GM1-/GD1a-), (wild-type CHO-K1 cells, GM1 and GD1a negative), and CHO-K1(GM1+/GD1a+) (genetically modified CHO-K1 cells that express both, GM1 and GD1a gangliosides) were probed with anti-GD1a monoclonal antibody or with cholera toxin (GM1) and analyzed by flow cytometry (**G**) and confocal microscopy (**H**).

**Figure 4 ijms-23-10354-f004:**
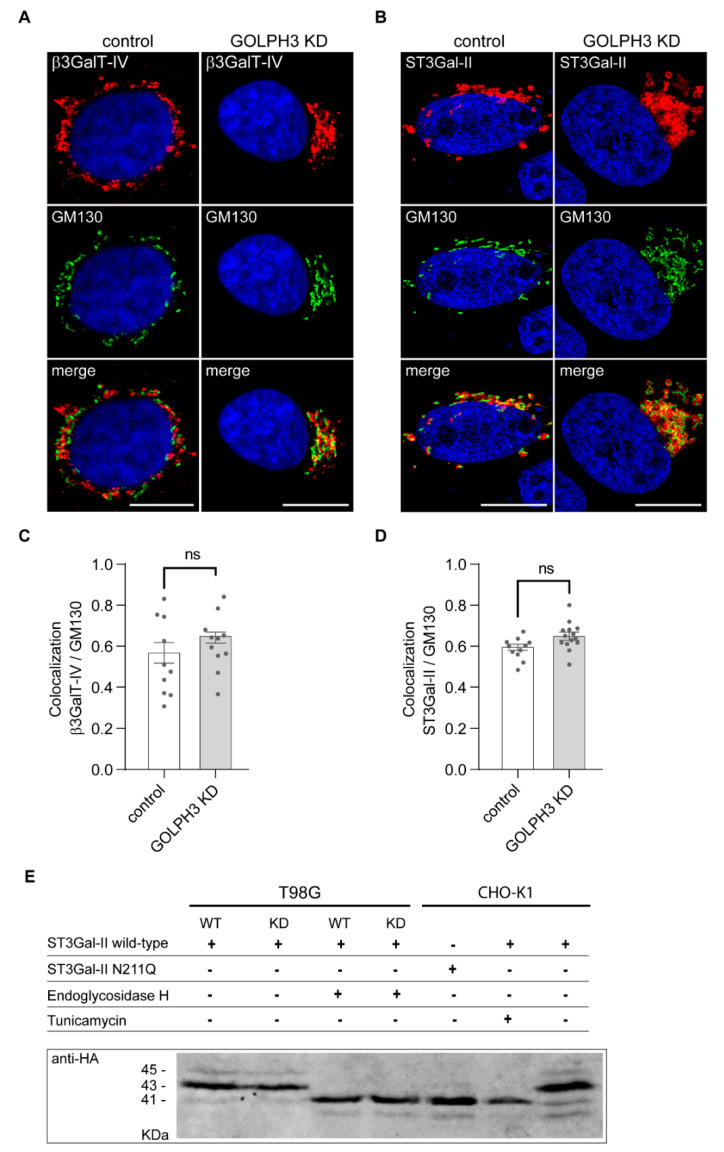
(**A,B**) Immunostaining showing the subcellular localization of β3GalT-IV and ST3Gal-II, respectively. and its colocalization with GM130 (130 kDa cis-Golgi matrix protein). Scale bars: 10 µm. (**C**,**D**) Quantification of the colocalization between GM130 with β3GalT-IV or ST3Gal-II, respectively. Two-tailed, unpaired t tests were carried out to assess statistical significance of results (ns: not significant). (**E**) Homogenates from T98G control and GOLPH3 knockdown (KD) cells transiently expressing a full-length, HA-tagged version of ST3Gal-II were treated (+) or not (−) with endoglycosidase H and subjected to SDS-PAGE and Western blot. As control, CHO-K1 cells were included in the analysis. CHO-K1 cells were transfected to express the wild-type, full-length, HA-tagged version of ST3Gal-II in the presence (+) or absence (−) of tunicamycin or an N-glycosylation mutant variant of the enzyme (ST3Gal-II N211Q). The expected molecular weight of unglycosylated and N-glycosylated ST3Gal-II is 41 and 43 kDa, respectively.

**Figure 5 ijms-23-10354-f005:**
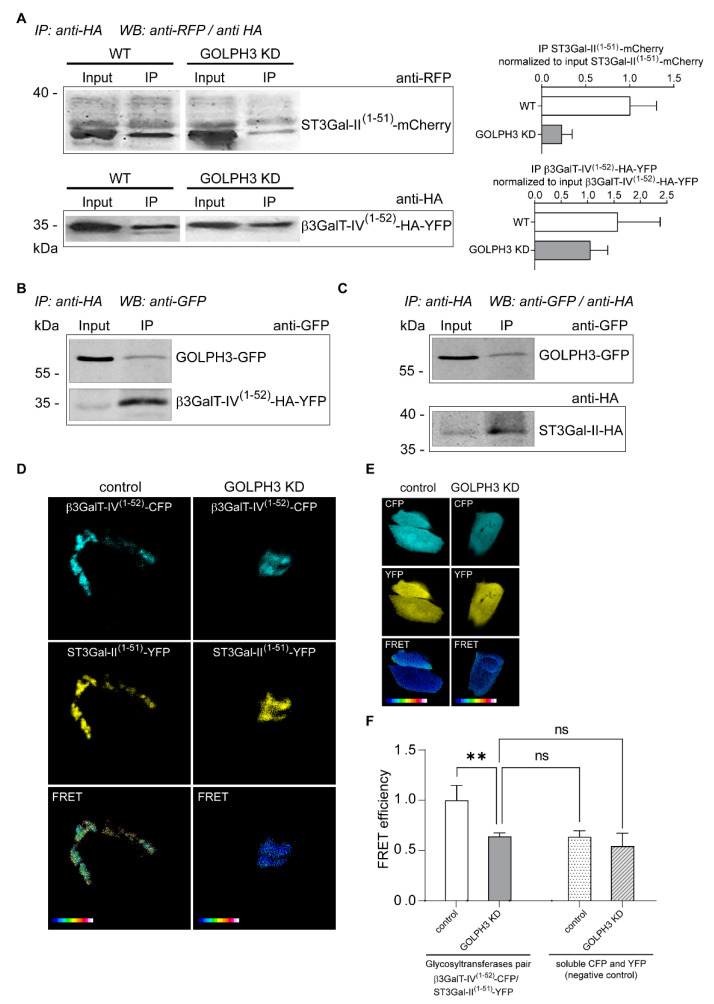
The physical association between ST3Gal-II and β3GalT-IV depends on GOLPH3. (**A**) Co-immunoprecipitation (IP) of lysates from control or GOLPH3 knockdown cells transiently expressing β3GalT-IV^(1−52)^-HA-YFP and ST3Gal-II^(1−51)^-mCherry was performed using anti-HA mAb as indicated under materials and methods. Immunocomplexes were analyzed by SDS-PAGE and Western blotting using polyclonal anti-RFP (ST3Gal-II(1-51)-mCherry) or anti-HA (β3GalT-IV(1-52)-HA-YFP) antibodies. A 10% of the total initial lysate (input) was included in each run. (**B**,**C**) Immunoprecipitation (IP) of lysates from T98G cells transiently expressing GOLPH3-GFP and β3GalT-IV^(1−52)^-HA-YFP (**B**) or GOLPH3-GFP and ST3Gal-II-HA (**C**) was performed by anti-HA antibody as indicated under materials and methods. Immunocomplexes were analyzed by SDS-PAGE and Western blotting using anti-GFP (recognizing both GFP and YFP in panel B and GFP in panel C) or anti-HA antibodies(recognizing HA-tagged ST3Gal-II in panel C). This anti-HA antibody was also used in a reblotting of the same membrane presented in panel B as an additional labeling control of β3GalT-IV(1-52)-HA-YFP (not shown). In total, 10% of the total initial lysate (input) was included in each run. IP indicates the antibody used for the immunoprecipitation whereas WB indicates the antibody used to develop the Western blot. (**D**) In vivo assessment of the occurrence of ST3Gal-II/β3GalT-IV complex by Förster Resonance Energy Transfer (FRET) analysis. Experiments were carried out in T98G control and GOLPH3 KD cells coexpressing the YFP and CFP versions of ST3Gal-II and β3GalT-IV, respectively, or the YFP and CFP (non-interacting proteins) proteins (**E**). Representative images of the FRET efficiency are shown. Scale ranges from white (maximum FRET efficiency) to blue (minimum FRET efficiency). (**F**) Graphic representation and statistical analysis of FRET results are shown. Results are mean +/− SEM of three independent experiments. Two-tailed, unpaired t tests were carried out to assess statistical significance of results (** *p* < 0.01; ns: not significant).

## Data Availability

The data presented in this study are available on request from the corresponding author.

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
