# Peer review of "Golgi Phosphoprotein 3 Regulates the Physical Association of Glycolipid Glycosyltransferases†"

_ijms, 2022, doi:10.3390/ijms231810354_

Round 1
Reviewer 1 Report
The current manuscript describes a nice work in the glycobiology field. Results are of interest to a large audience and the work deserves to be reported as it is.
Author Response
请参阅附件

Reviewer 2 Report
This manuscript by Ruggiero and coworkers described the possible formation of ternary complex between GOLPH3, b3GalT-IV, and ST3Gal-II. While compelling evidence for the formation of binary complex between b3GalT-IV and ST3Gal-II was observed through the co-immunoprecipitation experiments and the FRET techniques, the nature of interaction between b3GalT-IV and ST3Gal-II N-terminus was not clear. Chemistry-wise, it would be beneficial to confirm if such interaction was covalent-based or H-bonding-based. A careful examination of such binary complex under different pH conditions may provide the nature of bonding interaction.
The relative binding affinities between three components (GOLPH3, b3GalT-IV, and ST3Gal-II) using FRET experiments may also provide deeper understanding of the ternary complex proposed in the current manuscript.
Given the importance of oncoprotein GOLPH3, the current finding of possible ternary complex formation with b3GalT-IV and ST3Gal-II provides important information regarding the enzyme interaction changes in the surface expression pattern of gangliosides in cancer cells. While more work is needed to understand the assembly mechanism of ternary complex between GOLPH3, b3GalT-IV, and ST3Gal-II, the current studies provided strong evidence for the strong relevance of GOLPH3 to the GD1 and GM1 levels and activities.
In summary, this reviewer recommends the acceptance of the revised manuscript after the authors adequately address the above critiques.
Suggestions:
In Abstract---Several features such as...but the formation of ST3Gal-II/b3GalT-IV complex...
Author Response
请参阅附件。

Reviewer 3 Report
The manuscript entitled: “Golgi Phosphoprotein 3 Regulates The Physical Association of Glycolipid Glycosyltransferases” is really well written and interesting. The Introduction and the Discussion are intriguing and exhaustive, data are well presented and explained. All the information that should be presented are correctly reported (replicability and reproducibility, for instance).
I appreciate the quality of immunofluorescence and the analysis of distribution of the Golgi apparatus on xy planes.
In addition to some minor points that could be easily fixed, my concerns regard exclusively the last part of the manuscript in which the Authors report the first indication of physical interaction between ST3Gal-II and β3GalT-IV, which relies on GOLPH3. As this is quite an important point, I reported below some points that should be fixed/explained by the Authors. In particular, I am not convinced by the IP and Co-IP they show, also because the original western blot data attached to the manuscript are misleading.
Furthermore, I suggest the Authors to shorten the References section which is way too long.
Major points
- As regards the Figure 5A the yield of immunoprecipitation of b3-GalT-IV-HA-YFP is really poor considering that the input loaded is 10% of the total lysate. Usually, the IP of the bait should be at least abundant as the input. Given that this is a crucial point in the manuscript (as the Authors state that this is the first evidence of a direct interaction between the two enzymes) I recommend to strength the data shown. Have the Authors tested different binding conditions or beads? In my experience, HA-tagged beads work tremendously better than HA-Ab + agarose/sepharose beads. If the Authors are not able to repeat the Co-IP experiment, they could provide a quantification of the three experiment they have already performed. This could help to have a clear readout of the data. Moreover, they should provide a western blot showing the efficiency of GOLPH3-KD.
In the Figure legend of Figure 5, the panel A should be indicated as “co-immunoprecipitation” and not “immunoprecipitation”.
- Why the Authors use anti-RFP antibody instead that anti-mCherry? It is true that mCherry is a RFP, but there are many commercial anti-mCherry antibodies that would be more specific than a generic anti-RFP.
- The Original western blot data are extremely confusing for me in the way they are presented. For For both Figure 5B and for Figure 5C, the heading on the left reports that cells were transfected with “GalT2-HA-YFP and GOLPH3-GFP”, while the indication of the antibodies used on the right reports “b3-GalT-IV-HA-YFP” for Figure 5B and “GalT2-HA-YFP” for figure 5C, but I do see the same signal in the left part of the blot. Are those two different hybridizations of the same blot?
To avoid confusion, Authors must provide a new file with boxes indicating clearly which areas they cropped and used in the manuscript and below each box which antibody they used.
Anyway, Authors should provide in the Original western blot data the very same images used in the manuscript; the exposure of the raw Figure 5C is completely different from that used in the manuscript, as the IP signal for GOLPH3 is completely absent in the raw image, while clearly visible in the text.
Minor points
- Considering the massive Golgi rearrangement shown in T98G cells upon GOLPH-3 KD, did the Authors perform other functional studies to verify the overall functionality of the Golgi apparatus in addition to the alteration of ganglioside expression? Have the Authors observed alteration in the Trans-Golgi-Network?
- Have the Authors performed experiments (even not included in the manuscript) with drugs that change the organization of the Golgi apparatus (monensin or Brefeldin A) to compare the effects between drug treatment and GOLPH3-KD?
- What happens to the Golgi in cells other than T98G? In MCF7 cells, for instance, no Golgi marker is provided in the confocal fluorescence images. Does GOLPH3-KD alter Golgi distribution in cells (like MCF7) in which Golgi is located nearby the nucleus and not around it?
- The quality of GOLPH3 immunoblot in Figure 2 should be improved; it is difficult to perform densitometry on those images. The graph in Figure 2b shows relative expression in Fold, while in the text Authors report a reduction of GOLPH3 expression of 90%; I would change either the text or the graph to be consistent.
- Why the Authors report “homogenates” in the Figure legend of Figure 4? Did the Authors performed something different than total cell lysis? Could Authors comment on the band lower than 41 kDa which appears upon treatment with Endoglycosidase H in T98G cells and in steady state condition in CHO-K1 cells?
In the panel E there is no indication of the antibody used in the western blot (perhaps anti-HA?). There is a typo in the penultimate line of the figure legend: “Expeted”.
Round 2
Reviewer 2 Report
The authors' responses to this reviewer's critiques should be incorporated into the relevant sections of revised manuscript, in particular the discussion regarding the nature of interaction between b3GalT-IV and ST3Gal-II N-terminus (as in the response letter to the Reviewer).
The relative binding affinity between three components should be discussed in the revised manuscript at least the theoretical possibility, if not performed experimentally, using the relevant references.
Reviewer 3 Report
This reviewer is completely satisfied by Authors response. I appreciate the modifications and corrections they made in the manuscript and also the explanations they provided to my concerns.
